# Theranostic Risk Stratification for Thyroid Cancer in the Genomic Paradigm

**DOI:** 10.3390/cancers16081585

**Published:** 2024-04-20

**Authors:** Seza A. Gulec, Evander Meneses

**Affiliations:** 1Miami Cancer Research Center, Miami, FL 33181, USA; evander.meneses@hcahealthcare.com; 2Herbert Wertheim College of Medicine, Florida International University, Miami, FL 33199, USA

**Keywords:** theranostics, radioactive iodine theranostics, differentiated thyroid cancer, mis-differentiated thyroid cancer, radioactive iodine refractory, radioactive iodine indifferent, redifferentiation, genomics, molecular theranostics, risk stratification

## Abstract

**Simple Summary:**

Risk stratification for differentiated thyroid cancer is a well-established tool for prognostication and selection of the appropriate initial therapy for “differentiated thyroid cancers”. The current risk stratification systems include a system used in current guidelines, based on surgical pathology features identified postoperatively and a more dynamic model that includes all perioperative clinical, radiologic, and pathologic data as well as the response to initial therapy information. Theranostic risk stratification is essentially based on genomic and molecular features of the patient’s individual tumor, investigated perioperatively and associated or correlated with radioactive iodine theranostics. This approach allows a more rational selection of the extent of the initial surgical treatment and subsequent radioactive iodine treatment.

**Abstract:**

Theranostics define diagnostic evaluations directing patient-specific therapeutic decisions. Molecular theranostics involves genomic, transcriptomic, proteomic, metabolomic and finally phenonic definitions thyroid cancer differentiation. It is the functional differentiation that determines the sensitivity and accuracy of RAI imaging as well as the effectiveness of RAI treatment. Total thyroidectomy is performed to empower an anticipated RAI treatment. A preoperative determination of the genomic and transcriptomic profile of the tumor is a strong predictor of response to therapeutic interventions. This article discusses the oncopathophysiologic basis of the theranostic risk stratification approach.

## 1. Prologue

The practice of oncology has entered a personalized medicine paradigm with molecular theranostics. Cancer oncobiology is now defined based on genomic and epigenomic expressions. A clinical catalogue for genetic alterations associated with cancer has been created using next-generation sequencing and bioinformatics. The approach to clinical issues, thus, has moved beyond conventional diagnostics and therapeutic interventions. The molecular landscape of thyroid cancer is fairly well elucidated. In this new paradigm of genomics and molecular pathology, the risk stratification systems for thyroid cancer, based on traditional parameters, are challenged. The standard clinico-pathologic indicators of risk are gradually vacating their role to clear molecular markers [1].

## 2. Clinical Goals and Models for Risk Stratification

Risk stratification, like in all malignancies, is paramount in the management of thyroid cancer. The basic objective of risk stratification for thyroid cancer is to obtain prognostic information. The American Joint Committee on Cancer/tumor node metastasis (AJCC/TNM) staging guide is an established system that is used to predict disease-specific mortality [2]. Unlike many cancers, the risk of recurrence does not parallel mortality in differentiated thyroid cancer, the risk of recurrence far exceeding the risk of disease-specific mortality [3]. The staging systems designed to predict mortality in thyroid cancer are not predictive of disease recurrence. To address this issue, a three-tier risk stratification model was developed and first instituted in the ATA 2009 guidelines [3]. This model was further adapted in the ATA 2015 guidelines [4]. Although initially conceived as a three-category model of risk assessment [low, intermediate, or high risk], the ATA risk stratification system is now visualized as a continuum of risk, ranging from very low to very high risk of disease recurrence. This model is largely based on clinico-pathologic data provided in the surgical pathology appraisal. The risk stratification models of the ATA guidelines, though clinically useful to predict disease-specific mortality or overall survival, is suboptimal for long-term outcome predictions for an individual patient. A dynamic model was introduced by Tuttle et al. [5]. This model defined a process where the initial treatment stratification (based on preoperative and postoperative data) was modified over time as new data (response to treatment) became available. The dynamic risk stratification model adds a response-to-therapy re-evaluation and has more predictive value. The integration of response-to-therapy re-evaluations to the initial risk assessments provides more reliable outcome predictors.

A theranostic risk stratification system, beyond being predictive of disease-specific survival and the risk of recurrence, offers an evidence-based process that can govern treatment decisions, specifically for, but not limited to, the extent of initial surgical treatment and appropriate utilization of radioactive iodine (RAI). Currently, a “high-risk” disease designation typically triggers a strategy to “maximize” therapeutic interventions. The ideal strategy, however, should be *optimizing*, but not necessarily “maximizing” the strategies. Neither “less is more” nor “more for more” motto is appropriate. Theranostic risk stratification is patient-specific and is based on the elucidation of molecular markers that predict the therapeutic role and power of the RAI treatment, which is intimately linked to the indication and the clinical value of total thyroidectomy.

## 3. Theranostic Risk Stratification Model in Thyroid Cancer Management

Theranostic risk stratification is perioperatively initiated and dynamically updated throughout the clinical follow-up [6]. Perioperative risk stratification is a composite term referring to assembling patient-specific disease information to include preoperative, intra-operative, and postoperative data. In the perioperative, dynamic, theranostic model, the risk stratification information is extracted from preoperative imaging, cytology, molecular profiling, surgical exploration, surgical pathology, and postoperative imaging. In the theranostic model, the risk stratification begins prior to surgical treatment, with the molecular profiling of a suspicious nodule. Tumor biology information encrypted in the molecular profile determines the extent of initial surgical treatment (lobectomy vs. total thyroidectomy) beyond the size criterion, which clearly is far from being adequate. It is the molecular profile that ascribes a theranostic value to the nodule work-up. Clinical outcomes are directly linked to the mutational profile and the composition of the transcriptomic alterations. The effect of patient age and tumor size on the prognosis could merely be the reflection of a protracted time frame that allows for the accumulation of genomic and molecular alterations. The identification of clinico-pathologic or molecular predictors of recurrence or mortality does not and should not translate into a need to maximize interventions such as performing more extensive operations, expanding indications of RAI treatment, rigorous TSH suppression, or targeted therapies, but to select the appropriate interventions in the appropriate sequence.

Molecular theranostics defines a new paradigm in thyroid cancer risk stratification. New classification schemes, based on genomics and its phenotypic expressions are being formulated. Genomics with molecular pathology and molecular imaging reflect the true biologic nature of the different cancer types currently defined by conventional morphologic features. The tumor differentiation/de-differentiation and clinical behavior for each individual cancer are now being defined by molecular markers, in addition to standard morpho-pathology. Since the initiation of the cancer genome program in 2006, a large number of cancer types have been molecularly characterized under the Cancer Genome Atlas (TCGA) project [7]. In 2014 the first comprehensive study on a genomic characterization of thyroid cancer was published, compiling a large volume of data on morphologic and molecular features of papillary thyroid cancer [8]. “The integrated genomic characterization of papillary thyroid carcinoma” study marks the beginning of the new paradigm for thyroid cancer diagnosis and management. This study elucidated the pathways of thyroid cancer onco-physiology, and their impact on iodine metabolism. Correlations between morphology and driver genetic mutations as well as thyroid differentiation were first clearly described in a systematic fashion with this study (Figure 1). It since has become obvious that the traditional postoperative risk stratification criteria are insufficient to resolve the “***equipoise***” over the indications of total thyroidectomy and appropriate use of RAI treatment. There are always selection biases involved in the design of retrospective studies and the statistical “underpowering” or “overpowering” in the data analysis for prospective trials. The core matter, in truth, is the biological and functional heterogeneity of the “differentiated” thyroid cancers.

### 3.1. Mis-Differentiated Thyroid Cancer

The term “well-differentiated” thyroid cancer was originally intended to refer to a distinct morphologic architecture and nuclear morphology, but not to imply a functional differentiation. The term was introduced to the literature by Selvyn Taylor in 1962, to stress the significant differences in the clinical course of undifferentiated thyroid cancer and the differentiated varieties (papillary and follicular patterns) of thyroid cancer [9]. This convenient, yet overly simplistic classification changed the philosophy of thyroid cancer treatment drastically. The term “differentiated” was adopted to indicate a functional attribute, thus leading to a plausible conclusion that all “differentiated” thyroid cancers can most effectively be treated with RAI. Evidently, the high degree of variation in transcriptomic expressions for so-called differentiated thyroid cancers were not known at the time. The theranostic power of RAI is in fact dependent on the full expression of genes of iodine metabolism, involved in uptake, organification, and transportation.

The 2022 WHO classification of thyroid neoplasms stratify follicular cell-derived cancers based on molecular profiles and aggressiveness [10]. This classification aims to identify, diagnose, and group thyroid carcinomas from a clinical outcome and prognosis perspective. A major problem with this approach is that it continues to propagate the *misuse* of the term “differentiated” in the context of function and theranostics. The term “differentiated”, in fact, should be used more judiciously to indicate distinct morphologic types and subtypes where the follicular architecture is preserved to a degree. The interpretation of the molecular data in the context of functional differentiation and theranostics indicate that the majority of PTCs and some of the FTCs are “*mis-differentiated*”.

### 3.2. RAI-Indifference and RAI-Refractoriness

The mis-differentiated thyroid cancers have variable degrees of depressed iodine transcriptome and metabolomics depending on the genomic signature. The “mis-differentiated” cancers exhibit variable degrees of “***RAI-indifference***”. “Indifference” implies a lack of avidity or responsiveness to engage in RAI processing by the malignant tissue. Simply, the malignant tissue is not metabolically equipped to take up and process RAI as much as its non-neoplastic counterpart. The theranostic power of RAI is significantly diminished, the restoration of which requires a modulation of molecular pathways.

The functional “mis-differentiation” is a consequence of the constitutive activation of the MAPK signal transduction pathway due to oncoprotein mutations. The value of theranostic risk stratification is to connect the clinical efficacy of RAI molecular imaging and therapy to the cancer’s transcriptomics and metabolomics. The transcriptional, translational, and post-translational regulatory mechanisms of thyroid oncogenesis and their impact on morphologic and functional differentiation have been largely characterized. Molecular theranostics is a surrogate for RAI theranostics. The link involves the MAPK pathway signaling. The term “RAI-refractoriness” is a clinical term that defines biologically RAI-indifferent mis-differentiated thyroid cancers. A “***redifferentiation***” can be attained by the modulation of the MAPK pathway [11,12,13,14,15,16].

## 4. MAPK Signal Dysregulation and Functional De-Differentiation

PTC and FTC are driven by oncoproteins that signal, for the most part, through the MAPK pathway, though other signaling pathways such as PI3K/AKT are also associated with the oncogenesis of thyroid cancers. The MAPK signal dysregulation and feedback control mechanism constitute the biologic basis of differences in the oncogenic progression and therapeutic manipulations of the two most common mutations, i.e., BRAFv600E and RAS [6]. Oncocytic carcinomas have a different genomic profile. They are characterized by copy-number alterations and mitochondrial DNA mutations [8,17]. The poorly differentiated and undifferentiated cancers arise from papillary and follicular cancers as a result of a sequential accumulation of genetic mutations and transcriptomic and metabolic aberrations [18]. A constitutive activation of the MAPK pathway results in an enhanced ERK output. The ERK-mediated transcriptional program disrupts follicular morphologic differentiation and interrupts the expression of genes associated with thyroid functional differentiation. Different driver mutations are associated with different histologic variants of papillary thyroid carcinoma and confer distinct patterns of gene expression, signaling, and clinical characteristics [19,20,21]. BRAF-mutated classical or tall-cell-variant papillary thyroid carcinomas have a dampened RAI metabolism.

MAPK pathway dysregulation and feedback control are different in BRAF vs. RAS initiated tumors. Those driven by BRAFV600E do not respond to the negative feedback from ERK to RAF, resulting in high MAPK signaling [20]. RAS-driven tumors, on the other hand, signaling via RAF dimers, do respond to ERK feedback, resulting in a lower MAPK output. This differential signaling results in profound phenotypic differences. The expression of genes responsible for iodine uptake and metabolism are greatly reduced in BRAFV600E tumors, in contrast to the “RAF-dimer” tumors, and the expression of these genes is preserved to a degree. The oncobiology, oncopathophysiology of tumor progression, and therapeutic responsiveness to MAPK pathway modulation strategies are categorically different for these two primary drivers [21].

## 5. Thyroid Differentiation Score (TDS) and Decreased RAI Theranostic Power

TCGA studied the expression levels of 16 genes associated with thyroid metabolism and function designated as the TDS, which articulates a correlation between the mutational status of the tumors and their differentiation state (Figure 2). TDS show a strong correlation with the BRAF-RAS Score. The RAS-like PTCs have relatively high TDS values. The *BRAF*-like PTCs, however, show a wide range of TDS values, maintaining a consistent correlation with the BRAF-RAS Score, albeit to a lesser degree. TDS is also associated with architectural changes and correlates with tumor grade and prognostic risk, as expressed with the MACIS clinical risk score [8]. TDS is an integrated quantity conveying the relative expression of iodine-handling proteins. However, the TDS, as defined in the TCGA study, cannot be used for theranostic purposes because the patient gene expressions are compared to a diseased cohort median. In a study performed at our research center, we evaluated the fold change (FC) of mRNA expressions of thyroid-specific proteins between thyroid tumor and normal thyroid tissue. Our results (unpublished data) demonstrated more than 2-fold depression in the transcriptome of genes involved in the RAI theranostic circuit. Developing a TDS-theranostic (TDS-T) based on molecular cytology may have an impact on clinical decision making as to the extent of thyroidectomy and postoperative RAI therapy.

There is a significant variation in transcriptomics, proteomics, and metabolomics in patients with BRAFV600E mutated tumors. This may account for the range in functional and biological differentiation observed in this category and may explain the uncertainty regarding the prognostic, predictive, and theranostic power of the BRAFV600E mutation. Papillary mis-differentiation is associated with functional de-differentiation and a dampening of the theranostic power of RAI. The BRAF-RAS score has a predictive value that is not a precise indicator of the theranostic power of RAI. The development of a theranostic TDS score has a strong potential to become a predictive measure for the theranostic power of RAI.

The theranostic risk stratification model assigns preoperative molecular profiling a critical role for initial treatment planning. In the theranostic paradigm, a total thyroidectomy decision is not independent of the preoperative molecular profiling. A molecular profile indicating a low TDS may not be amenable to an “effective” adjuvant RAI treatment. This group, thus, may not benefit from total thyroidectomy with or without RAI treatment, unless the MAPK cascade is blocked. Current schemas of blockage involve MEK inhibitors, BRAF inhibitors, and the combination of both [12,13,14]. BRAF-induced MAPK activation cannot be controlled with MEK inhibition only and requires combined MEK and BRAF inhibition, whereas the activation can be controlled reasonably well in RAS mutation-initiated cancers [15,16,22]. Oncocytic carcinomas (OCs), formerly named Hürthle cell cancers, and considered a variant of follicular thyroid carcinoma, are shown to be a genomically distinct entity and classified as a separate cancer type [19]. OCs are traditionally considered to be refractory to radioactive iodine. The transcriptomic profile of OCs shows significant heterogeneity. Most OCs, particularly those with aggressive biology and clinical courses, have a low TDS. The molecular mechanisms connecting the primary genomic aberrations in OCs to a depressed TDS profile are not clear. Whether the depressed TDS can be reverted in OCs is open to investigation.

The true index of functional differentiation is an orderly preservation of iodine metabolic machinery–iodine uptake to organification. This function can be interrogated at genomic/transcriptomic (next-generation sequencing), proteomic (immunohistochemistry), metabolomic (autoradiography), and finally phenomic (RAI imaging) levels (Figure 3). Functional differentiation can be quantitated in vivo and in vitro by the TDS, that is, determined by a molecular analysis and by RAI imaging after a complete removal of thyroid gland. The TDS is a transcriptomic surrogate for the theranostic power of RAI and can be determined by molecular cytology. This signifies a paradigmatic change in initial surgical treatment planning for thyroid cancers.

## 6. The Role for Molecular Theranostics in Determining the Extent of Initial Surgical and RAI Treatments

Total thyroidectomy is performed to facilitate the subsequent RAI treatment. The dynamic–theranostic stratification system includes molecular markers identified preoperatively. One could then evaluate the potential value/benefit of RAI on an individual basis and plan a “coupled” surgery–RAI treatment strategy. This new system revives the surgery–RAI coupling based on clear clinical indications of RAI treatment derived from molecular predictors as obtained from preoperative molecular/genomic profiling of the nodules and from imaging performed in the postoperative and post-RAI treatment period.

The traditional initial treatment protocol advocates total thyroidectomy for high-risk (larger than 4 cm) tumors, and adjuvant RAI treatment for high-risk cancers based on surgical pathology findings. The coupling of surgery and RAI treatments is based on the assumption that all “differentiated” thyroid cancers have adequate RAI avidity. The impact of initial surgical treatment to overall outcome of differentiated thyroid cancer in an independent (uncoupled) role is much less clear than its collective value with RAI treatment. In all reality, surgical treatment is tightly coupled with postoperative RAI treatment. Total thyroidectomy does not have an intrinsic power to improve clinical outcomes. This is best demonstrated with two powerful studies reported by the MSKCC group 30 years apart [23,24]. Other large-scale retrospective studies that are frequently quoted in the equipoise years have significant shortcomings in their design and analyses [25,26,27]. To reemphasize, the main purpose of total thyroidectomy is to remove the functioning thyroid tissue to divert the RAI toward the malignant tissue with depressed RAI avidity [28]. This RAI biokinetics-based initial treatment strategy, over the years, lost its original purpose, and total thyroidectomy became a standard procedure for the management of thyroid cancer. In the last decade, in an attempt to better define the role for total thyroidectomy–RAI treatment, prognostic risk stratification systems were developed. As a result, the patients who were deemed to have high-risk disease were recommended to receive the most intense treatment schedule. However, a major oversight is that it is this group of patients who have the most profound thyroid functional depression and cannot possibly directly benefit from the RAI treatment other than the remnant ablation intent. In the high-risk group, total thyroidectomy–RAI treatment planning should include a redifferentiation strategy.

The traditional paradigm was built upon vigilant clinical data analysis without having the benefit of molecular and genomic information. There is a bias towards more aggressive treatments combining surgery and RAI treatments based on deductive reasoning and retrospective data. The principles of RAI treatment were first systematized for clinical practice by Bierwaltes [29,30] and further refined by Mazzaferri, who generously contributed to the thyroid cancer literature with his meticulous analyses of retrospective data [25,31]. The Beirwaltes–Mazzaferri paradigm dominated the field including the ATA 2009 guidelines. The second period is represented in the ATA guidelines of 2015 and is marked by a trend to a “highly selective” use of RAI treatment based on a postoperative risk assessment model. The ATA low-, intermediate-, and high-risk categories were defined. RAI treatment was discouraged in the low-risk category and largely reserved for the high-risk category. The intermediate category was left in equipoise. Ironically, what was clearly demonstrated by the thyroid cancer genomics data was that the true high-risk “differentiated” thyroid cancers were functionally not differentiated.

## 7. Molecular Cytology and Theranostics

It is imperative to identify the indications for the potential to enhance the efficacy of RAI treatment prior to the initial surgical treatment. An appropriate choice of initial surgical treatment option is adherent to the a priori determination of the potential value of RAI treatment. This includes the value of RAI both as an ablative tool and its adjuvant power. The advances in the oncobiologic characterization of thyroid nodules and thyroid cancer allow one to make these deductions and decisions. A major advance in risk stratification is the implementation of the “perioperative dynamic–theranostic model”.

Molecular theranostics has the potential to identify different disease categories than the traditional clinico-pathologic diagnostics. A minimalistic approach may be appropriate for papillary microcarcinomas, with a low-risk molecular profile. Nodules identified as such can be addressed with local ablative procedures or may even be safely followed using a deferred intervention [active surveillance] plan. On the other end of the spectrum, those papillary microcarcinomas with high-risk molecular profiles, may warrant a total thyroidectomy. The complete surgical removal of disease may not necessarily require total thyroidectomy. There is no clear evidence that total thyroidectomy, in the absence of nodal disease and tumor confined to one lobe and without the adjuvant benefit of radioactive iodine treatment, improves disease outcome. Complete surgical removal of the thyroid gland, however, is a prerequisite for a radioactive iodine treatment to prevent normal thyroid tissue from diverting RAI away from the neoplastic tissue. Surgical treatment, therefore, is tightly coupled with the radioactive iodine treatment potential. When the initial choice is a lobectomy, there is no role for radioactive iodine treatment in the management unless surgical pathology of the lobectomy specimen indicated risk factors that obviated a completion of the thyroidectomy. For total thyroidectomy to have a therapeutic benefit, the tumor molecular profile should allow an adequate response to RAI. Those tumors with unfavorable molecular profile for RAI may need a molecular therapeutic modulation prior to surgery–RAI treatment plan.

## 8. Epilogue

“Completeness” in oncologic surgery is defined as an R0 resection. Clinically apparent nodal metastases are addressed with therapeutic lymph node dissection. If an RAI treatment for ablation or adjuvant purposes is not planned preoperatively, a total thyroidectomy may not be required for an R0 resection, even for tumors >4 cm. Total thyroidectomy is surgically indicated when anatomically, there is contralateral lobe involvement, and oncologically, when a benefit from an RAI treatment is anticipated. Complete thyroidectomy, in the most precise sense, can only be accomplished by surgical total/near-total thyroidectomy followed by RAI treatment. One should be prepared to deal with structural artefacts originating from remnant tissue identified by ultrasound or RAI imaging during the surveillance and elevated but diagnostically inconclusive thyroglobulin (Tg) levels. The incidence of regional lymph node (LN) metastatic involvement and the fate [natural course] of the “latent” metastatic lymph nodes are not known. Although the regional nodal basin represents the most common site of disease recurrence, a prophylactic lymph node dissection has never been shown to have an impact on clinical outcomes.

RAI treatment following total or near-total thyroidectomy is performed with three intentions: (1) for the elimination of normal thyroid remnant; (2) for the coverage of presumed occult (or better termed as synchronous metastatic/latent disease, typically nodal) disease, and (3) for the treatment of known residual or metastatic disease (therapy). The terms remnant ablation, adjuvant treatment, and therapy are used for these specific indications, respectively. Theranostic risk stratification plays a role in guiding the latter two. The ablation of a thyroid remnant is merely performed to facilitate Tg follow-up and by the elimination of potential structural artefacts to allow easier surveillance with anatomic and functional imaging. RAI remnant ablation completes surgical total thyroidectomy.

MAPK pathway modulation to revert the indifference to RAI prior to an RAI treatment in both the adjuvant treatment setting and the treatment for metastatic disease is emerging as a new strategy [32]. It can be regarded as a “neoadjuvant” intervention to render occult or overt disease responsive to RAI therapy. The theranostic power of RAI can be improved with new and evolving redifferentiation strategies.

## 9. Conclusions

Existing risk stratification systems can and should be refined, by the incorporation of patient- and tumor-specific molecular markers that have theranostic value, to optimize patient-specific (individualized) treatment decisions. Appropriate interventions can be employed in an appropriate sequence. A limited number of molecular markers (mutations) have been included in the ATA risk stratification model, but their theranostic value has not been explored. A full integration of molecular theranostics in a risk stratification model is needed to improve patient care. An ideal risk stratification model should provide tumor biology data and connect these biologic data to therapeutic decisions, serving as a theranostic instrument.

## Figures and Tables

**Figure 1 cancers-16-01585-f001:**
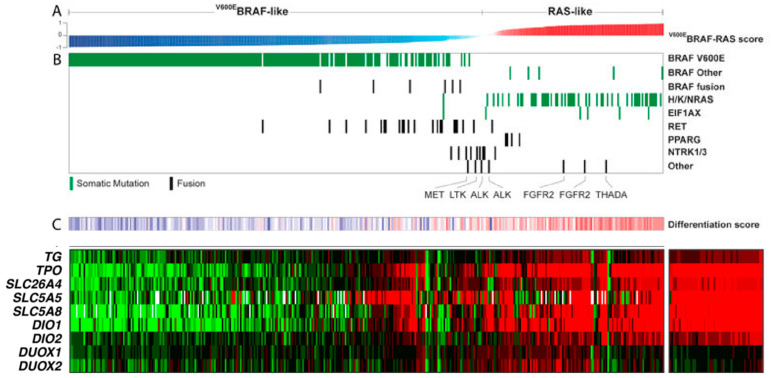
Genomic and transcriptomic layout of papillary thyroid cancer and their correlations with the thyroid differentiation score (TDS) and clinical risk stratification (modified from [8]). (**A**) The TCGA study developed a BRAF-RAS score to quantify an individual tumor’s gene expression profile. The numerical scale of this score is a range between [−1] and [+1]. BRAF-like PTCs are in negative and RAS-like PTCs are in positive polar direction with a strong separation of the BRAFV600E- and RAS-mutant tumors. (**B**) The expression profiles of the other less common mutations have been tabulated on the BRAF-RAS scale. (**C**) TDS and RAI theranostic transcriptome pattern correlates with BRAF-RAS score. The theranostic transcriptome is depicted as a heat map. Green and red colors indicate depressed and preserved gene expressions respectively.

**Figure 2 cancers-16-01585-f002:**
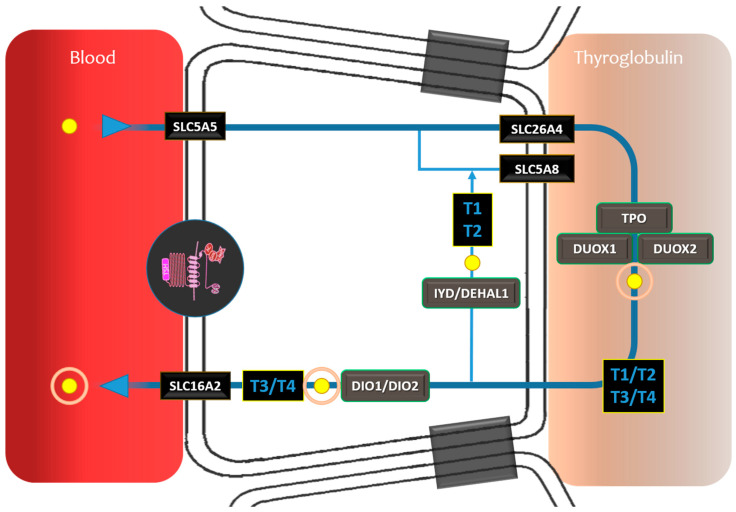
RAI theranostic circuit involving all the steps from uptake, transport, organification, and thyroid hormone release. Gene expressions critical for theranostic TDS. SLC5A5 (solute carrier family 5 (sodium/iodide cotransporter), member 5): sodium–iodide symporter activity, responsible for the uptake of iodine in the thyroid. SLC26A4 (solute carrier family 26 (anion exchanger), member 4): iodide transmembrane transporter activity. TPO (thyroid stimulating hormone receptor): iodination of tyrosine residues in thyroglobulin and phenoxy-ester formation between pairs of iodinated tyrosines to generate the thyroid hormones, thyroxine, and triiodothyronine. DUX01 (dual oxidase 1): involved in the synthesis of thyroid hormone. DUX02 (dual oxidase 2): involved in the synthesis of thyroid hormone. DIO1 (deiodinase, iodothyronine, type I): activates thyroid hormone by converting the prohormone thyroxine (T4) by outer ring deiodination (ORD) to bioactive 3,3′,5-triiodothyronine (T3). DIO2 (deiodinase, iodothyronine, type II): activates thyroid hormone by converting the prohormone thyroxine (T4) by outer ring deiodination (ORD) to bioactive 3,3′,5-triiodothyronine (T3). IYD (iodotyrosine deiodinase): encodes an enzyme that catalyzes the oxidative NADPH-dependent deiodination of mono- and diiodotyrosine, which are the halogenated byproducts of thyroid hormone production. SLC5A8 (solute carrier family 5 (sodium/monocarboxylate cotransporter), member 8): transport iodide by a passive mechanism. SLC16A2 (solute carrier family 16 member 2): encoded protein facilitates the cellular importation of thyroxine (T4), triiodothyronine (T3), reverse triiodothyronine (rT3), and diidothyronine (T2). TG (thyroglobulin): substrate for the synthesis of thyroxine and triiodothyronine as well as the storage of the inactive forms of thyroid hormone and iodine. TSHR (thyroid stimulating hormone receptor): receptor for thyrothropin and a major controller of thyroid cell metabolism.

**Figure 3 cancers-16-01585-f003:**
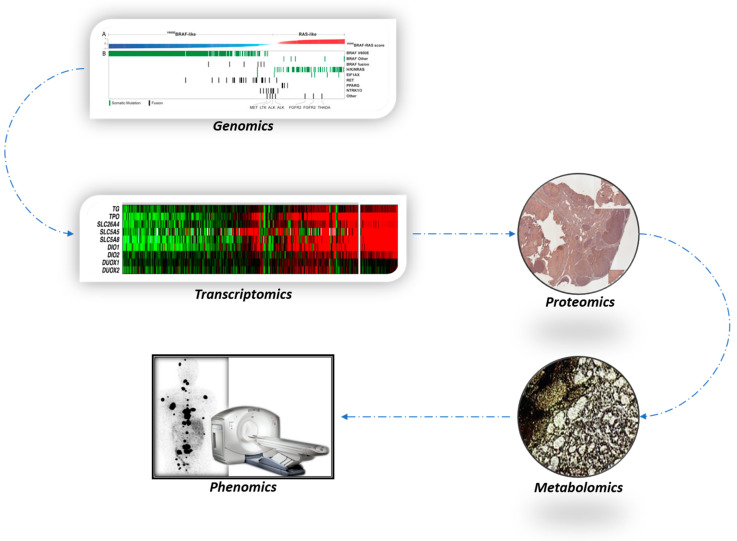
Genomics is predictive of transcriptomics, and transcriptomics is predictive of proteomics. Both genomics and transcriptomics can be conveniently determined by next-generation sequencing of fine needle aspiration biopsy material. Proteomics, by immunohistochemistry, demonstrates tissue expression and a functional orientation of the determinants of thyroid differentiation. RAI metabolomics implies the distribution pattern of RAI in normal and malignant neoplastic tissue. The suppressed RAI uptake can be demonstrated by autoradiography. This technique is currently not in practical clinical use. The in-vivo ultimate theranostic power of RAI is expressed as phenomics and most decisively determined by RAI imaging and dosimetry.

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
