# Peer review of "Theranostic Risk Stratification for Thyroid Cancer in the Genomic Paradigm"

_cancers, 2024, doi:10.3390/cancers16081585_

Round 1
Reviewer 1 Report
Comments and Suggestions for Authors
1. Whenever mentioned the word “significant” – should also reporting the corresponding p-value and tests done to support such statement.
2. Should the reference notation font format be superscript style?
3. Figure 1 correlation – is there a statistical tests done to report the degree of correlation?
4. Table 1 should add a title to describe the contents and the purpose of such table.
5. Suggested to add a section mention the limitation of the method/study.
Reviewer 2 Report
Comments and Suggestions for Authors
Dear authors,
The manuscript “THERANOSTIC RISK STRATIFICATON FOR THYROID CANCER IN THE GENOMIC PARADIGM”, cancers-2942132, reviews known staging systems and guidelines for thyroid carcinoma patient management and discusses the therapeutic role and power of RAI treatment. The paper suggests the use of a new theranostic risk stratification system for thyroid carcinoma treatment selection. The paper is written in an easy-to-follow way, contains some interesting observations, and with a few minor corrections, I recommend it for publication.
1. Line 23-25: The reference is missing. Please add it.
2. Line 44: The description of the RAI abbreviation is missing. Please add it.
3. Figure 1: Figure 1's description is poor. The words and abbreviations used in the figure (such as TDS, risk, description of colors) should all be defined beneath the figure. In addition, do the terms "risk" in the figure and "BRS" signify the same thing?
4. Lines 132-133: The reference is missing. Please add it.
5. Lines 156-165: These lines can also be seen in Figure 1. I would recommend mentioning Figure 1 at some point in this paragraph.
6. Line 169: I would advise you to place Figure 1 in the brackets (as the TDS are indicated in Figure 1, not in Table 1), and then you could add a line explaining what is displayed in Table 1. Furthermore, Table 1's name is absent. Kindly include it. Alternatively, as Table 1 is not discussed much in the article, it could only be removed.
7. The explanation of TDS abbreviation should be added to the Figure 1 description and should not be repeated (lines 166, 169, 188, 214).
8. Line 306: The description of the LN abbreviation is missing. Please add it.
Reviewer 3 Report
Comments and Suggestions for Authors
1) In this study, the authors aim to review theragnostic risk stratification for thyroid cancer in the genomic paradigm in the view of models of risk stratification, MAPK signal activation and thyroid de-differentiation. While the content of this manuscript is superficial, lacking clear organization in its discussion, and the synthesis of thyroid cancer risk stratification is unsatisfactory.
2) Line 47-50: “Theragnostic risk stratification is patient-specific…the clinical value of total thyroidectomy”. Please discuss in detail.
3) Introducing the concept of the thyroid differentiation score is indeed meaningful. However, this article neglects to address how it intersects with the risk stratification outlined by the author.
4) The figures in the text all have a black background, which is not recommended.
5) Apart from abnormalities in the MAPK signal pathway, pathways such as PI3K/Akt are also associated with the prognosis of differentiated thyroid cancer.
Comments on the Quality of English LanguageNo.
Round 2
Reviewer 3 Report
Comments and Suggestions for Authors
The manuscript has undergone some improvements after modifications, but there are still some issues remaining with its quality
1. Figure 3: Please refer to more relevant articles concerning iodine uptake, process, and RAI. Also, please improve the aesthetics of the image as it is too simplistic.
2. This is a review article; however, upon thorough examination, the molecular mechanisms of thyroid tumors are not adequately comprehensive and in-depth. It takes a lot of time and thoughtful reflection to summarize!! The following papers are recommended for reference: 1) Genomic alterations in thyroid cancer: biological and clinical insights; 2) Pathogenesis of cancers derived from thyroid follicular cells; 3) Single-Cell Transcriptome Analysis Reveals Inter-Tumor Heterogeneity in Bilateral Papillary Thyroid Carcinoma; 4) Thyroid cancer (Lancet, 2023)...
3. Is Figure 4 based on the author's own research findings? If not, please provide the source and reference.
Comments on the Quality of English LanguageNo comments.
